# Evaluation of Different Capture Solutions for Ammonia Recovery in Suspended Gas Permeable Membrane Systems

**DOI:** 10.3390/membranes12060572

**Published:** 2022-05-31

**Authors:** María Soto-Herranz, Mercedes Sánchez-Báscones, Juan Manuel Antolín-Rodríguez, Pablo Martín-Ramos

**Affiliations:** 1Departamento de Ciencias Agroforestales, ETSIIAA, Universidad de Valladolid, Avenida de Madrid 44, 34004 Palencia, Spain; mercedes.sanchez@uva.es (M.S.-B.); juanmanuel.antolin@uva.es (J.M.A.-R.); 2Instituto Universitario de Investigación en Ciencias Ambientales de Aragón (IUCA), EPS, Universidad de Zaragoza, Carretera de Cuarte s/n, 22071 Huesca, Spain; pmr@unizar.es

**Keywords:** ammonia adsorbents, ammonia recovery, citric acid, organic acids, phosphoric acid, water

## Abstract

Gas permeable membranes (GPM) are a promising technology for the capture and recovery of ammonia (NH_3_). The work presented herein assessed the impact of the capture solution and temperature on NH_3_ recovery for suspended GPM systems, evaluating at a laboratory scale the performance of eight different trapping solutions (water and sulfuric, phosphoric, nitric, carbonic, carbonic, acetic, citric, and maleic acids) at 25 and 2 °C. At 25 °C, the highest NH_3_ capture efficiency was achieved using strong acids (87% and 77% for sulfuric and nitric acid, respectively), followed by citric and phosphoric acid (65%) and water (62%). However, a remarkable improvement was observed for phosphoric acid (+15%), citric acid (+16%), maleic acid (+22%), and water (+12%) when the capture solution was at 2 °C. The economic analysis showed that water would be the cheapest option at any working temperature, with costs of 2.13 and 2.52 €/g N (vs. 3.33 and 3.43 €/g N for sulfuric acid) in the winter and summer scenarios, respectively. As for phosphoric and citric acid, they could be promising NH_3_ trapping solutions in the winter months, with associated costs of 3.20 and 3.96 €/g N, respectively. Based on capture performance and economic and environmental considerations, the reported findings support that water, phosphoric acid, and citric acid can be viable alternatives to the strong acids commonly used as NH_3_ adsorbents in these systems.

## 1. Introduction

The application of manure as a fertilizer in areas with a high concentration of livestock can lead to a surplus of nutrients in the soil [1] and damage ecosystem quality, causing eutrophication of surface and groundwater bodies, soil acidification, and global warming [2]. In the European Union (EU), the livestock sector is estimated to contribute to 78% of biodiversity loss, 73% of N and P water pollution, 80% of soil acidification and air pollution (NH_3_ and NO_x_ emissions), and 81% of global warming caused by the whole agricultural sector [3]. In response to this situation, the EU has promoted policies to improve the utilization of manure nutrients in agriculture and reduce their environmental impact through the Nitrates Directive (Directive 91/676/EEC EC) and the National Emission reduction Commitments Directive (Directive (EU) 2016/2284). In the framework of meeting the objectives set in these policies, and due to the high price of fertilizers, there is a growing interest in developing technologies to reduce emissions of NH_3_ and to recover total ammonia nitrogen (TAN) from animal manure as a concentrated fertilizer product [4]. This would allow the production of nitrogen fertilizers that can compete with conventional synthetic fertilizers produced by the Haber-Bosch process, contributing to the achievement of circular economy goals [5].

A wide variety of technologies are currently available for TAN recovery, such as air scrubbers [6], reverse osmosis [7], adsorbent materials [8], evaporation techniques [9], struvite precipitation [10], nitrification-denitrification processes [11], the anaerobic ammonium oxidation (anammox) process [12], microbial fuel cells [13], or treatment with microalgae and photosynthetic bacteria [14,15]. However, they have important limitations that hinder their widespread application: air scrubbers and zeolite adsorption techniques require manure pre-treatment [16], reverse osmosis requires high working pressures, struvite precipitation requires additives [17], and biological treatments are only effective at low to medium TAN concentrations [18], given that bacterial activity can be inhibited in the presence of high organic matter and nitrogen contents [19].

In recent years, gas-permeable membrane (GPM) technology has emerged as an alternative to the above-mentioned approaches. This technology, based on the passage of NH_3_ through a hydrophobic microporous membrane and its subsequent capture and concentration in an extraction solution on the other side of the membrane, offers advantages such as high transfer surface area, low working pressures, no need for manure clarification pre-treatment or additives [16], low energy consumption, and the possibility of being used in combination with other treatment technologies [20]. Its main drawback is the cost of installation and maintenance of the membranes [18].

Laboratory and pilot-scale studies have demonstrated the suitability of GPM technology based on expanded polytetrafluoroethylene (e-PTFE) membranes for N recovery from liquid effluents [16,21], wastewater from food industries [22], or sewage sludge [23]. Promising results have also been reported for the capture of NH_3_ volatilized from animal manures [24,25,26].

Concerning the optimization of the capture process, issues such as the effects of pH, temperature, and airflow on the recovery efficiency of TAN from manure have been discussed in the literature [27,28]. However, the impact on process performance associated with the use of extraction solutions alternative to sulfuric acid (normally used to recover NH_3_ in the form of (NH_4_)_2_SO_4_, despite its safety and environmental risks) has seldom been analyzed in such previous research. Some authors have explored the possibility of using other mineral acids (nitric acid, phosphoric acid, acid mixtures, etc.) [29,30,31,32] for NH_3_ removal from tertiary effluents, wastewater, or human urine, while other researchers have used organic acids, carbonic acid, or water [33,34,35] for NH_3_ recovery from anaerobic sewage sludge digestate, landfill leachate, or wastewater. Nonetheless, the afore mentioned studies explored those alternative capture solutions in air stripping systems, liquid-liquid membrane contactors, hollow fiber membrane contactors, scrubbing systems, flow-electrode capacitive deionization systems, or capacitive membrane extraction systems (CapAmm), not on air-suspended GPM systems.

In view of this research gap, the present study aimed to investigate—on a laboratory scale—the NH_3_ capture efficiency of different extraction solutions in a suspended GPM system. Specifically, a comparison in terms of performance was made between all the stripping solutions proposed in the above-referred studies (viz. sulfuric, phosphoric, nitric, carbonic, acetic, citric, and maleic acids, and deionized water). As a secondary objective, the influence of the temperature of the capture solution on NH_3_ recovery was analyzed, an important aspect for the application of this technology in real livestock facilities conditions that had not been addressed in previous works. Thirdly, the operating costs, together with the associated chemical expenditure, were estimated for each trapping solution. The reported technical and preliminary economic evaluation may be useful not only to researchers working on more sustainable membrane-based systems but also to livestock farm owners and agricultural operators interested in producing certified organic fertilizers (which is not possible if mineral acids are used).

## 2. Materials and Methods

### 2.1. Reagents

Synthetic ammonia solutions were prepared with 18.2 MΩ·cm Milli-Q water (Merck Millipore, Burlington, MA, USA). Other reagents used were: sulfuric acid (CAS 7664-93-9, 98%), phosphoric acid (CAS 7664-38-2, 85%), nitric acid (CAS 7697-37-2, 65%), acetic acid (CAS 64-19-7, 99.7%), citric acid (CAS 77-92-9, 99.5%), maleic acid (CAS 110-16-7, 99%), and N-allylthiourea (CAS 109-57-9, 98%). All chemicals were analytical grade reagents and were supplied by Panreac Química S.L.U. (Barcelona, Spain).

### 2.2. Experimental Conditions

The experimental setup was described in detail in previous work by our group [36]. The diagrams of the setups used for the NH_3_ capture assays performed at 25 °C, at 2° C, and using carbonic acid are presented in Appendix A, respectively.

It should be clarified that the selected temperatures for the capture solutions (25 and 2 °C) correspond to the average temperatures, during the period 1999–2019, of the warmest month (August) and the coldest month (January) in the town of Santa María la Real de Nieva (Segovia, Spain), where it is planned to carry out pilot-scale tests in the near future. Given that capture solution tanks are installed outside the livestock buildings, these would be the approximate temperatures at which the capture solutions would be in summer and winter.

To minimize variability resulting from the use of manures with inconstant NH_3_ concentrations, a 6000 mg NH_3_-N·L^−1^ synthetic solution was used in all the tests, consisting of 24.6 g NH_4_Cl·L^−1^ + 43.2 g NaHCO_3_·L^−1^ + 10 mg N-allylthiourea∙L^−1^ (as a nitrification inhibitor). The pH of the synthetic solutions was kept above 8 in all experiments (similar to those of real emitting sources such as pig slurry, chicken manure, laying hen manure, etc. [16,20,37] and the temperature was kept at 25 °C (which is the usual setpoint temperature in many Spanish farms [38]), to replicate real farm conditions. It is worth noting that the pH affects the TAN (NH_4_^+^/NH_3_) equilibrium, in such a way that values above 8 promote the conversion of NH_4_^+^ to NH_3_, resulting in a higher presence of free ammonia and favoring mass transfer through the membrane [39].

Any capture solution can be used in the NH_3_ extraction process as long as its pH is below the value of the acidic dissociation constant of the ammonium/ammonia pair (pKa = 9.24), where pK is the logarithm of the reciprocal of the acidic dissociation constant (Ka). Therefore, for all NH_3_ capture solutions tested, it was necessary to keep the pH below this pKa value [32,40]. However, in this study, pH corrections with concentrated acid were not necessary in any case (except in the case of carbonated water traps, where CO_2_ was dosed at a pressure of 0.1 bar depending on the pH present in the medium (pH < 6.36)).

The 1N capture solutions were continuously circulated inside the membrane using a Pumpdrive 5001 peristaltic pump (Heidolph, Schwabach, Germany) at a flow rate of 2.1 L·h^−1^.

The absorption surface (163.4 cm^2^) was the same in all experiments, and the characteristics of the suspended e-PTFE membrane at the beginning of the experiment (supplied by Zeus Industrial Products Inc., Orangeburg, SC, USA) are summarized in Appendix A. Given that the membranes were not reused, a morphological analysis of the membranes was not conducted after the 7-day experiments.

For each capture solution-temperature combination, three runs were carried out over 7 days.

### 2.3. Analysis Methodology

During the experiments, pH, electrical conductivity, temperature, and NH_3_-N concentration of the capture solutions and the synthetic N-emitting solution were monitored. pH, electrical conductivity, and temperature were measured with a GLP22 electrode (Crison Instruments S.A., Barcelona, Spain). NH_3_-N concentration was determined by steam distillation, subsequent collection of distillates in borate buffer, and titration with 0.2 mol·L^−1^ HCl. A Kjeltec 8100 apparatus (Foss Iberia S.A., Barcelona, Spain) was used for distillation.

### 2.4. Data Calculations

The amount of NH_3_-N emitted (expressed in mg NH_3_-N) was determined by the difference between the amount of NH_3_-N present at the beginning and the end of the experiment in the synthetic N-emitting solution. The mass of NH_3_-N recovered (in mg NH_3_-N) was determined as the amount of NH_3_-N present at the end of the experiment in the capture solution. To calculate the NH_3_-N capture efficiency of the different extraction solutions (NH_3_-N efficiency, %), the mass of NH_3_-N recovered was divided by the amount of NH_3_-N emitted. The overall ammonia removal efficiency was calculated by comparing the amount of NH_3_-N recovered in the trapping solution with the initial NH_3_-N concentration of the emitting solution. Finally, the NH_3_-N mass flux across the membrane (in mg NH_3_-N·cm^−2^·d^−1^), produced as a consequence of the gas concentration gradient across the membrane, was estimated by considering the average mass of N captured per day and the membrane surface area.

### 2.5. Statistical Analysis

First, the data were tested for homogeneity and homoscedasticity using the Shapiro-Wilk and Levene tests. A two-way analysis of variance (ANOVA) was then conducted, followed by a post hoc comparison of means using Tukey’s test (*p* < 0.05). R statistical software [41] was used for the statistical analyses.

### 2.6. Economic Analysis of the Different NH_3_ Capture Solutions

To compare the different capture solutions used, an economic analysis was carried out for each of them. To this end, operation costs (associated with the electric power consumption of the capture solution recirculation pump) and chemical costs (quantity of acid necessary to carry out the captures obtained in this study or quantity of injected CO_2_) were considered, which are summarized in Appendix A for the summer and winter scenarios, respectively. Reagent prices were obtained from various chemical suppliers and should be taken as estimates, given that actual values fluctuate with the market.

In the experiments referred herein, which lasted 7 days, the saturation of the trapping solutions was not reached in any case, so the compensations offered by the final products obtained were not relevant and were not considered in the economic analysis.

## 3. Results and Discussion

### 3.1. pH and Electrical Conductivity Evolution of NH_3_ Trapping Solutions

The pH and electrical conductivity initial and final values (after 7 days) of the eight trapping solutions (sulfuric, phosphoric, nitric, carbonic, acetic, citric, and maleic acids, and water), at the two temperatures tested (25 and 2 °C), are shown in Table 1.

The pH of the trapping solutions increased in all cases at the end of the experiment, indicating that all solutions were capable of trapping NH_3_. The smallest differences between the pH measured at the end and the beginning of the experiment were found for the solutions based on strong acids (i.e., sulfuric and nitric), while the largest increases in pH occurred for water at both temperatures (6.7→8.6 and 6.0→8.7), acetic acid at 2 °C (2.2→3.8), citric acid at both temperatures (1.7→3.0) and phosphoric acid at both temperatures (1.3→2.0).

The increase in pH in the water-only trapping solution can be attributed to the high solubility of NH_3_ in water [42]. In the case of the 2 °C acetic acid trapping solution, the higher pH increase (compared to the 25 °C scenario) could be due to lower volatilization of acetic acid at lower temperatures, which would have favored NH_3_ capture [33].

As for the EC values in the different trapping solutions, a strong decrease was observed in the sulfuric and nitric acid solutions at the end of the experiment. In this regard, strong acids are totally ionized in an aqueous solution, showing a very high EC and yielding many H^+^ ions to react with NH_3_. As NH_3_ capture occurs, the conductivity of the solution decreases because the acidic permeate solution with high conductivity slowly changes to an NH_4_SO_4_/NH_4_NO_3_ solution (which has lower conductivity than the previous acidic medium) [32]. In the case of phosphoric acid, this behavior is not so marked because, although it is an inorganic acid, it has a weaker character.

In the case of weak organic acids, the conductivity remained stable or increased, as their starting conductivity is low and that of the formed salts is higher.

### 3.2. NH_3_-N Recovery Comparison at Different Temperatures

#### 3.2.1. Evolution of NH_3_-N Recovery

Figure 1 shows the evolution of the amount of NH_3_-N recovered by the eight trapping solutions over the 7-day experimental period at the two temperatures (25 and 2 °C). 

The accumulation of NH_3_-N in the capture solutions followed a linear trend at both temperatures, with R^2^ values above 0.93 and 0.91 at 25 and 2 °C, respectively, consistent with the behavior observed in other studies for sulfuric acid-based trapping solutions conducted over 14-day and 240-day periods [26,36].

At 25 °C, the average capture rates were 397 ± 12, 295 ± 43, 352 ± 20, 49 ± 1, 25 ± 13, 295 ± 53, 252 ± 12, and 285 ± 41 mg NH_3_-N·d^− 1^ for sulfuric acid, phosphoric acid, nitric acid, carbonic acid, acetic acid, citric acid, maleic acid, and water capture solutions, respectively. At 2 °C, the capture rates were 407 ± 19, 362 ± 27, 376 ± 17, 53 ± 5, 238 ± 25, 369 ± 34, 354 ± 34, and 338 ± 12 mg NH_3_-N·d^− 1^ for sulfuric acid, phosphoric acid, nitric acid, carbonic acid, acetic acid, citric acid, maleic acid, and chilled water, respectively.

#### 3.2.2. NH_3_-N Recovery Comparison at 25 °C

Figure 2 shows the total amount of NH_3_-N recovered by the eight trapping solutions at the two temperatures (25 and 2 °C) at the end of the experiment.

When the trapping solutions were at 25 °C (summer scenario), the sulfuric and nitric acid solutions showed the best response in terms of NH_3_ trapping, followed by phosphoric acid, citric acid, deionized water, maleic acid, carbonic acid, and acetic acid.

Such higher recovery observed for strong inorganic acids is due to the stronger interaction between ammonium and the anion resulting from acid dissociation [32], and the results are consistent with those obtained by Damtie et al. [32], who used hydrophobic hollow fiber membrane contactors to recover NH_3_ from human urine at room temperature, using sulfuric acid, nitric acid, phosphoric acid, and water (as a control) as capture solutions. They found that sulfuric acid was slightly better at NH_3_ capture than other acids, but without significant differences. In this regard, it should be clarified that in our study there were no statistically significant differences between sulfuric acid and nitric acid, but there were significant differences between sulfuric acid and phosphoric acid. Such differences may be tentatively attributed to the fact that ammonium phosphate is less soluble in water than ammonium sulfate, with solubility values of 42.9 kg·L^−1^ for [(NH_4_)_2_PO_4_] and 70.6 for [(NH_4_)_2_SO_4_] at 0 °C, and of 68.6 vs. 75.4 kg·L^−1^ at 20 °C, respectively [31].

Concerning the organic acids, the NH_3_ trapping results at 25 °C may be deemed as good (except for acetic acid): citric acid was significantly different from sulfuric acid, but not from nitric and phosphoric acids, and maleic acid was also comparable to phosphoric acid. It is worth noting that citric acid has three H^+^ ions per acid molecule and is stronger than the other two assayed organic acids, which explains why better NH_3_ capture results were obtained. As for maleic acid, authors such as Starmans and Melse [43] agree that, among the organic acids, it is the second-best alternative to sulfuric acid after citric acid.

The observed behavior differs from that reported by Jamaludin et al. [33] for an anaerobic digestate NH_3_ stripping system using citric and acetic acids, cold water, and mineral salts (Epsom and gypsum). These authors found that citric acid achieved a similar stripping performance to sulfuric acid (with an NH_3_ recovery of 561.3 mg versus 607.5 mg, respectively), but at the cost of using twice as much citric acid as sulfuric acid (due to its partial dissociation characteristics). In this study, the NH_3_ recovery for both solutions was significantly different (2065 mg for citric acid versus 2776 mg for sulfuric acid).

As for the low NH_3_ recovery attained using acetic acid as the trapping solution, it may be due to its lower acid strength (*K*_a_ = 1.75 × 10^−5^) compared to citric (7.45 × 10^−4^) or maleic (9.1 × 10^−3^) acids [33], and to its high Henry’s volatility constant (*K*_H_ = 2.50 × 10^−2^ m^3^ Pa mol^−1^). This is >10^12^ times higher than, for example, those of sulfuric (7.69 × 10^−14^ m^3^ Pa mol^−1^) or citric acid (3.33 × 10^−17^ m^3^ Pa mol^−1^) [44]. Using acetic acid, Jamaludin et al. [33] obtained recoveries of 204.3, 421, and 536.1 mg NH_3_ for scrubbing temperatures of 67–70, 36–40, and 15–18 °C, respectively, compared to 561.3 mg NH_3_ when using citric acid.

In the solution generated by bubbling CO_2_ in water (carbonic acid), very low NH_3_ recovery values were also observed (343 mg NH_3_). This result can be tentatively attributed to the fact that the CO_2_ that is not absorbed in water, in the gas phase, can generate a competitive occupation of the membrane pores, reducing the NH_3_ flux inside the membrane, which in turn leads to a reduction of NH_3_ capture. This phenomenon was also observed by authors such as Zhang et al. [45] in CapAmm extraction systems using carbonated water as an adsorbent.

In relation to the water solution, NH_3_ amounts similar to those obtained with weak acids were recovered. In this sense, it must be taken into consideration that NH_3_ has a high solubility in water due to its polarity, forming hydrogen bonds with water molecules [46], and is retained in the solution provided that pH 9.2 is not exceeded [32,45]. However, the good NH_3_ uptake results can also be attributed to the joint diffusion of NH_3_ and CO_2_ through the membrane, in such a way that, in addition to ammonium hydroxide and NH_3_(aq), ammonium bicarbonate is also present in the uptake solution. Such presence of bicarbonate in the water samples was confirmed by laboratory analysis by titration with 0.1N HCl (not shown). This would be in good agreement with the results obtained by Vanotti et al. [47]: in a study on NH_3_ recovery from poultry litter, they found that the amount of nitrogen recovered in the tank after 37 days was higher using distilled water as the trapping solution than using 1N sulfuric acid (4012 vs. 1156 mg N·L^−1^, respectively). Based on analyses of NH_3_ and carbonate concentrations in the recirculated water, they showed that gaseous NH_3_ was recovered as ammonium bicarbonate salt and concluded that CO_2_ also permeated through the e-PTFE membrane (similar to the one used in this study) and drove NH_3_ fixation with water.

#### 3.2.3. NH_3_-N Recovery Comparison at 2 °C

NH_3_ recovery at 2 °C increased in all trapping solutions compared to the values reported above at 25 °C (Figure 2), although differences were only statistically significant in the case of acetic acid and maleic acid. Being aqueous solutions, the observed improvement may be mainly attributed to the increase in NH_3_ solubility in water with decreasing temperature [48] (ca. 90 g/100 mL at 0 °C vs. ca 32 g/100 mL at 25 °C). However, differences associated with the acid present in the capture solution cannot be ruled out. For instance, in the case of sulfuric acid, Kurtén et al. [49] reported that NH_3_:H_2_SO_4_ ratios of nucleating clusters varied as a function of temperature and NH_3_ concentration. At a 1 ppm NH_3_ concentration, they estimated that (H_2_SO_4_)_2_·NH_3_ clusters’ relative concentration would decrease from 50.2% to 4.2% when the temperature was decreased from 25 to 0 °C, whereas (H_2_SO_4_)_2_·(NH_3_)_2_ clusters’ relative concentration would increase from 49.8 to 95%. Hence, a higher capture ratio should be possible at 0 °C for the same amount of acid.

#### 3.2.4. Differences in NH_3_ Flux, NH_3_-N Capture, and NH_3_-N Removal Efficiencies

The NH_3_ flux rate, NH_3_-N capture efficiency, and overall NH_3_-N removal efficiency values for the eight capture solutions at 25 and 2 °C are summarized in Table 2.

Most of the calculated NH_3_ fluxes across the e-PTFE membrane were higher than those reported by Zhang et al. [45] (0.56–1.21 mg·cm^−2^·d^−1^), which should be mainly ascribed to the use of a much more concentrated NH_3_-emitting synthetic solution (43 mg NH_4_^+^-N vs. 6000 mg NH_3_-N·L^−1^). Soto-Herranz et al. [36] reported flux rates of 2.14 ± 0.2 mg N·cm^−2^·d^−1^ using sulfuric acid as the NH_3_ trapping solution at 25 °C, for an NH_3_ emitting solution concentration, membrane surface area, and flux rate similar to those used in this study, so the results reported herein (2.4 ± 0.1 mg N·cm^−2^·d^−1^) would be comparable.

The NH_3_ capture yield at 25 °C ranged from 62% (water) to 87% (sulfuric acid), except for acetic acid (6%) and carbonic acid (11%) solutions. As discussed above, this could be due to the volatile character of acetic acid [44] and the competitive nature of CO_2_ molecules in the membrane pores—which would impair NH_3_ transfer—, respectively.

In the winter scenario, at 2 °C, the NH_3_ capture efficiency barely changed for strong acids (+2% and +5% for sulfuric and nitric acid, respectively), but substantially improved for the other trapping solutions: phosphoric acid (64→79%), citric acid (65→81%), maleic acid (55→77%) and water (62→74%). Regarding acetic acid, although it significantly improved its capture yield up to 52%, it was still lower than that of the aforementioned capture solutions. As for the CO_2_ saturated solution (H_2_CO_3_), the yield continued to be very low (12%).

Comparing the values obtained with those reported in the literature, the NH_3_ capture efficiencies for sulfuric, phosphoric, and nitric acid obtained in this study were similar to those attained by other authors [40,45], and the differences may be attributed to the fact that the capture system was not the same. Riaño et al. [50], using a submerged GPM system with a sulfuric acid capture solution, reported capture efficiencies of up to 79.7%; Zhang et al. [45], using a CapAmm system, obtained NH_3_ capture efficiencies of 75% for phosphoric acid and 73% for sulfuric acid; and Reig et al. [40], using hollow fiber liquid-liquid membrane contactors (HF-LLMC) combined with ion exchange, attained NH_3_ capture efficiencies of 74% and 85% with nitric and phosphoric acid, respectively.

Using citric acid, Jamaludin et al. [33] obtained NH_3_ recoveries from digestate of up to 90%, higher than that achieved in this study at low temperature (81%).

On the other hand, the NH_3_ capture yields achieved using water as the capture solution (62–74%) were higher than the values referred by authors such as Jamaludin et al. [33] and Zhang et al. [45], who obtained efficiencies in the 35–61% range.

Using carbonic acid as the NH_3_ capture solution, recovery efficiencies of 11–12% were obtained, lower than those reported by Zhang et al. [45].

Concerning the overall NH_3_-N removal efficiencies, they were in the 34–48%, 29–43% and 33–39% range for mineral acids, citric and maleic acid, and water, respectively. These values were approximately half of the NH_3_-N capture efficiencies, given that suspended GPM systems only recover nitrogen from the headspace (i.e., the TAN retained in the emitting solution cannot be recovered) and the average NH_3_-N emission in the reported experiments was ca. 53%. For comparison purposes, in the work by Samani-Majd et al. [25], in which a hybrid GPM system (combining a submerged GPM system with a suspended GPM system) with a sulfuric acid trapping solution was used for ammonia recovery from free-stall dairy manure, an overall removal efficiency of 48% was reached; and in the work by Riaño et al. [50], focused on ammonia recovery from raw swine manure with a submerged GPM system using a sulfuric acid capture solution, the overall removal efficiency was in the 14.3–49.5% range.

### 3.3. Comparison of Economic Costs Associated with the Different NH_3_ Capture Solutions on a Laboratory Scale

The costs associated with the use of each capture solution in the summer (25 °C) and winter (2 °C) scenarios are shown in Figure 3.

In the summer scenario (25 °C), estimated operation costs would be lower for strong inorganic acids (1.82 and 2.05 €/g N for sulfuric acid and nitric acid, respectively) than for weak acids (2.44, 2.44, and 2.86 €/g N for phosphoric, citric, and maleic acid, respectively) or water (2.52 €/g N). Due to the low recovery yields, the operation costs for acetic acid and carbonic acid would be much higher (28.51 and 14.68 €/g N, respectively) and their use would not be advisable.

In the winter scenario (2 °C), operation costs would decrease in all cases, but not in the same proportion. Differences between those associated with the use of strong acids (1.77 and 1.91 €/g N for sulfuric and nitric, acid respectively) and those resulting from the weak acid-based alternatives (1.99, 1.95, and 2.04 €/g N for phosphoric, citric and maleic acid, respectively) or water (2.13 €/g N) would be noticeably smaller.

Zhang et al. [45] also estimated lower operating costs (1.02–1.30 €/kg N) when using inorganic acid sorbents (sulfuric, phosphoric, nitric, and hydrochloric acid) than weak sorbents (1.7 and 2.5 €/kg N for carbonic acid and water, respectively). However, the estimated costs are only comparable within each study, due to the differences in energy and reagent costs between the study by Zhang et al. [45] and the present study.

Regarding chemical costs, in the summer scenario (25 °C), those associated with the use of organic acids (2.52 and 3.52 €/g N for citric and maleic acid, respectively) would be noticeably higher than those of strong acids (1.61 and 2.04 €/g N for sulfuric and nitric acid, respectively). Interestingly, the use of phosphoric acid would be the cheapest option (1.49 €/g N), if water is excluded. These differences would again be substantially reduced in the winter scenario: while chemical costs reduction would be smaller than 6.5% for strong acids (1.57 and 1.91 €/g N for sulfuric and nitric acid, respectively), it would be in the 18.6–28.9% range for weak acids (1.21, 2.01, and 2.51 €/g N for phosphoric acid—again the cheapest option—, citric acid, and maleic acid, respectively). In the case of the H_2_CO_3_ solution, since ammonia extraction contributed to CO_2_ fixation, the expenditure on reagents could be considered negative (for instance, Zhang et al. [45] considered earnings of $ 0.37 kg^−1^ N).

If the total costs associated with the use of each trapping solution in both scenarios (summer and winter) are considered, the most viable option for NH_3_ capture in both cases would be water, with a total cost of 2.52 €/g N in summer and 2.13 €/g N in winter, a result consistent, for example, with that obtained by Zhang et al. [45] for a CapAmm system. Although the NH_3_ capture efficiency using this capture solution was not the highest, the savings in reagents make it a very interesting option as an NH_3_ capture solution.

Citric or phosphoric acid solutions would also be viable options for NH_3_ capture, with total associated costs of 3.96 and 3.20 €/g N, respectively (compared to 3.33 €/g N for sulfuric acid) in the winter scenario, and 4.96 and 3.93 €/g N, respectively (compared to 3.43 €/g N for sulfuric acid) in the summer scenario. The use of citric acid may be interesting to produce organic fertilizers suitable for organic farming [51], which cannot be produced from mineral acids. However, double-dosing would be required, due to its partial dissociation [33]. The phosphoric acid choice would be supported by economic considerations and safety considerations (in comparison with sulfuric and nitric acids).

Although in the study presented in Figure 3 the profit contributed by the final products obtained has not been included, it should be taken into consideration that, although the NH_3_ capture efficiencies with water (62–74%) were lower than those achieved by the rest of the capture solutions, the final product has a higher market value than those derived from other fertilizers such as sulfuric acid (25% ammoniacal solutions have a selling price of approximately 1.30 €/kg N, compared to 0.69 €/kg N for 21% ammonium sulfate salt). As for weak acid end products, market prices for (NH_3_)_2_HPO_4_ and ammonium acetate (ca. 1.10 and 1.06 €/kg N, respectively) are also higher than that of (NH_4_)_2_SO_4_. The additional revenue from these end products would make water and phosphoric acid trapping solutions even more attractive and could help to offset the higher costs associated with the use of citric acid.

### 3.4. Applicability and Limitations of the Study

#### 3.4.1. Applicability of the Study

The use of distilled (or tap) water as the capture solution would be the most recommendable option, as it has a moderate/high NH_3_ capture performance regardless of the operating temperature used and the associated costs would be the lowest. In this case, the final product obtained (ammonia water) could be used, for instance, in waste incinerators for NO_x_ capture by selective (non-catalytic) reduction [33].

Concerning the use of weak acids, as noted above, phosphoric acid would be cheaper than strong acids in winter conditions. However, despite being more expensive, the use of citric acid as an alternative capture solution may also be advantageous in winter. In this regard, there is consensus that there is a significant increase in NH_3_ emissions in the summer months compared to the winter months [52], as the increase in outside temperature in the summer months causes an increase in the temperature inside the farm, which accelerates the NH_3_ release rate from the slurry/manure [53] and increases the concentration of gases in the air. However, the above reasoning ignores the impact of ventilation. Guo et al. [54] studied the concentration ranges of air pollutants in pig and poultry houses equipped with forced ventilation systems and found that the concentrations greatly varied depending on the season: for pig houses, the NH_3_ concentration ranged from 27 ppm in winter to 0.8 ppm in summer, and for poultry houses, from 25 ppm in winter to 2 ppm in summer.

Since the capture efficiency is directly proportional to the NH_3_ concentration in the livestock housing environment, the winter period would be the most favorable to apply GPM technology. In fact, in a pilot-scale NH_3_ capture study using suspended GPM in pig and poultry houses [26], the NH_3_ capture efficiency rate in winter was 2 to 4 times higher than in summer. Taking this factor into consideration, and in view that the total cost associated with the use of citric acid trapping solutions in winter conditions would be 18.8% and 3.6% higher than those resulting from the use of strong acids (sulfuric and nitric acid, respectively), their replacement with this safer and more environmentally benign alternative would be supported. Furthermore, as indicated above, the resulting salt is considered a biofertilizer, which has a higher market value than salts formed from inorganic acids [33] and does not contain sulfur, making it suitable for soils with excess sulfur problems [55]. In addition, it improves the bioavailability of phosphorus in the soil for plant uptake, as citrate ions form complexes with metal cations, thus solubilizing precipitated and adsorbed phosphorus [56].

#### 3.4.2. Limitations of the Study

In this work, the performance of different solutions for NH_3_ gas capture at a laboratory scale was studied to select the most promising ones. However, under such strictly controlled working conditions, the reported capture performances are expected to be higher than those achievable under real farm conditions. In this regard, a previous study [26] showed that capture yields were reduced by the management practices of the farms where the pilot plant based on a suspended GPM system was installed, atmospheric conditions, etc. Therefore, studies under real conditions in different types of livestock housing facilities, locations, and over longer periods are necessary to confirm the preliminary conclusions obtained in the laboratory.

In addition, since the behavior of the capture solutions was only evaluated at two temperatures (2 and 25 °C), a more detailed investigation of the performance of the NH_3_ capture solutions over a wider range of temperatures and using smaller intervals would be needed to extrapolate the results to the varied atmospheric conditions that may occur in livestock facilities.

It is also important to note that the economic study presented in Section 3.3 has been carried out in an economic context marked by uncertainty and high price volatility (due to the conflict in Ukraine), so it is not possible to make cost comparisons with previous studies.

## 4. Conclusions

This study investigated the use of different trapping solutions at two temperatures (2 and 25 °C) in a GPM system with suspended e-PTFE membranes to recover NH_3_ released from a synthetic solution of 6000 mg N·L^−1^. At 25 °C (summer scenario), strong acids (sulfuric and nitric) were the most efficient, followed by weak acids (phosphoric, citric, and maleic) and water, with efficiencies of 87 and 77%, 55–65%, and 62%, respectively. Despite their advantages in terms of lower reagent costs, trapping solutions based on acetic acid and carbonated water (carbonic acid) had to be discarded due to the high volatility of the former and possible back-diffusion of CO_2_ and NH_3_ through the membrane when the latter was used. In winter conditions (2 °C), the NH_3_ capture efficiencies of all solutions improved, with particularly noticeable increases for phosphoric, citric, and maleic acids (+15%, +16%, and +22%, respectively). Bringing together technical and economic considerations, water may be put forward as the most advisable trapping solution, given that it offers moderate-high recovery yields in both summer and winter conditions and 24–38% lower total costs than traditionally used strong inorganic acids. Concerning weak acids, given that the capture efficiencies of GPM systems under real operating conditions are higher in winter and that cost differences would be small, it may also be advisable to opt for citric acid or phosphoric acid in winter. In addition to a lower environmental impact, the use of these three alternative capture solutions would have the advantage of obtaining end products with a higher market value, which would partly offset the costs associated with the implementation of GPM technology.

## Figures and Tables

**Figure 1 membranes-12-00572-f001:**
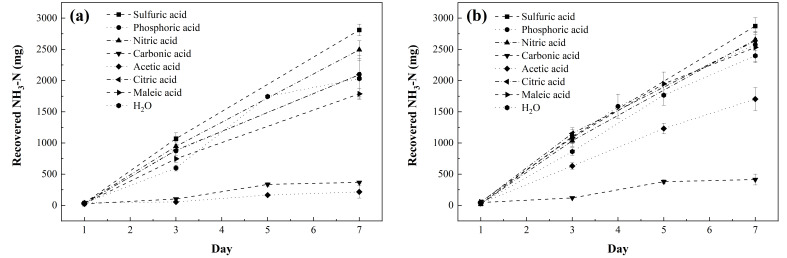
Evolution of the amount of TAN captured in the different trapping solutions during the 7-day experiments at (**a**) 25 °C and (**b**) 2 °C. All values are expressed as mean ± s.d. of n = 3.

**Figure 2 membranes-12-00572-f002:**
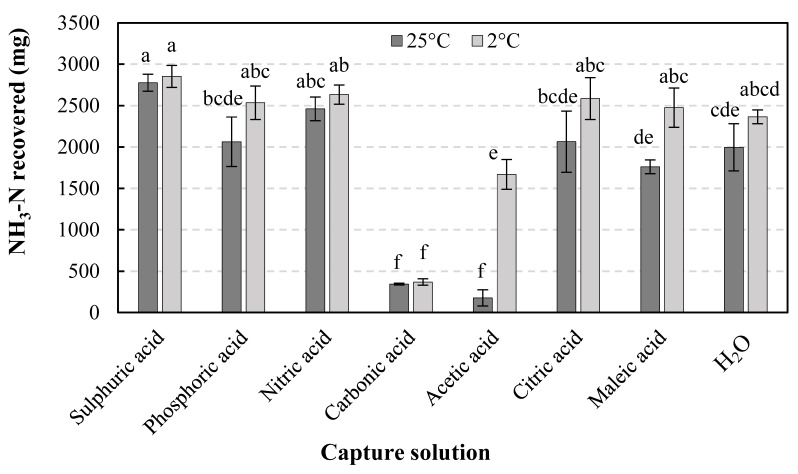
Mass of NH_3_-N recovered by the different trapping solutions at two temperatures after 7 days. Values with different letters are significantly different at *p* ≤ 0.05 according to Tukey’s HSD test. All values are expressed as means of n = 3.

**Figure 3 membranes-12-00572-f003:**
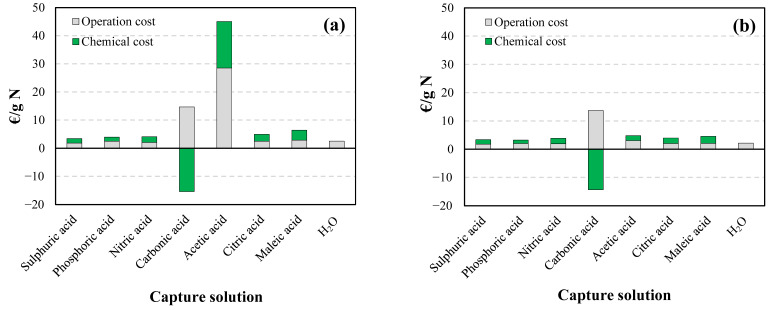
Economic analysis of the suspended GPM-based NH_3_ capture process using eight different types of trapping solutions in the (**a**) summer (25 °C) and (**b**) winter (2 °C) scenarios.

**Table 1 membranes-12-00572-t001:** pH and EC values at the beginning and the end of the experiment for the different NH_3_ trapping solutions at two temperatures, 25 and 2 °C, representative of the winter and summer scenarios, respectively.

Stripping Solution	Parameter	T (°C)	Experimental Time	
			Day 1	Day 7
Sulfuric Acid	pH	25	0.6 ± 0.0	0.7 ± 0.0
		2	0.3 ± 0.1	0.5 ± 0.2
	CE	25	207.3 ± 0.6	170.5 ± 3.1
		2	289.7 ± 19.3	206.7 ± 5.5
Phosporic Acid	pH	25	1.3 ± 0.1	2.0 ± 0.0
		2	1.3 ± 0.1	2.0 ± 0.1
	CE	25	23.2 ± 0.3	18.6 ± 0.4
		2	22.9 ± 0.6	18.2 ± 0.9
Nitric acid	pH	25	0.5 ± 0.0	0.6 ± 0.1
		2	0.4 ± 0.0	0.6 ± 0.0
	CE	25	224.7 ± 1.2	174.3 ± 14.5
		2	234.7 ± 1.2	181.2 ± 2.6
Carbonic acid	pH	25	6.0 ± 0.6	6.6 ± 0.0
		2	5.9 ± 1.3	6.5 ± 0.1
	CE	25	0.2 ± 0.0	4.4 ± 0.2
		2	0.3 ± 0.1	3.5 ± 0.3
Acetic acid	pH	25	2.4 ± 0.0	2.9 ± 0.2
		2	2.1 ± 0.1	3.8 ± 0.0
	CE	25	1.5 ± 0.1	1.9 ± 0.5
		2	1.6 ± 0.2	9.3 ± 0.6
Citric acid	pH	25	1.7 ± 0.1	3.0 ± 0.1
		2	1.7 ± 0.1	3.0 ± 0.0
	CE	25	5.8 ± 0.1	11.2 ± 1.0
		2	5.7 ± 0.1	12.3 ± 0.4
Maleic acid	pH	25	1.3 ± 0.0	1.5 ± 0.1
		2	1.1 ± 0.0	1.7 ± 0.1
	CE	25	31.9 ± 0.1	26.5 ± 0.1
		2	30.8 ± 0.3	25.2 ± 0.1
H_2_O	pH	25	6.7 ± 0.2	8.6 ± 0.1
		2	6.0 ± 0.2	8.7 ± 0.0
	CE	25	0.7 ± 0.1	12.4 ± 1.8
		2	0.4 ± 0.2	9.6 ± 0.1

All values are expressed as mean ± s.d. of n = 3.

**Table 2 membranes-12-00572-t002:** NH_3_ flux rates (N flux), NH_3_-N capture efficiencies, and overall NH_3_-N removal efficiencies for the different trapping solutions at the two assayed temperatures, 25 and 2 °C, representative of the winter and summer scenarios, respectively.

Capture Solution	T (°C)	N Flux (mg N·cm^−2^·d^−1^)	NH_3_-N Capture Efficiency (%)	NH_3_-N Removal Efficiency (%)
Sulfuric acid	25	2.4 ± 0.1 a	87	46
	2	2.5 ± 0.1 a	89	48
Phosphoric acid	25	1.8 ± 0.3 bcde	64	34
	2	2.2 ± 0.2 abc	79	42
Nitric acid	25	2.2 ± 0.1 abc	77	41
	2	2.3 ± 0.1 ab	82	44
Carbonic acid	25	0.3 ± 0.0 f	11	6
	2	0.3 ± 0.0 f	12	6
Acetic acid	25	0.2 ± 0.1 f	6	3
	2	1.5 ± 0.2 e	52	28
Citric acid	25	1.8 ± 0.3 bcde	65	34
	2	2.3 ± 0.2 abc	81	43
Maleic acid	25	1.5 ± 0.1 de	55	29
	2	2.2 ± 0.2 abc	77	41
H_2_O	25	1.7 ± 0.3 cde	62	33
	2	2.1 ± 0.1 abcd	74	39

Values followed by the same letter are not significantly different at *p* ≤ 0.05 according to Tukey’s HSD test. All values are expressed as mean ± s.d. of n = 3.

## Data Availability

The data presented in this study are available on request from the corresponding author. The data are not publicly available due to their relevance as part of an ongoing Ph.D. thesis.

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
