# Peer review of "Evaluation of Different Capture Solutions for Ammonia Recovery in Suspended Gas Permeable Membrane Systems"

_membranes, 2022, doi:10.3390/membranes12060572_

Round 1

Reviewer 1 Report

In the present study, the author reported the ammonia recovery performance of eight different capture solutions at two temperatures (25 and 2 °C) using suspended expanded GPMs in a laboratory scale experimental setup. Additionally, this manuscript also provided an economic evaluation for different ammonia capture solutions. However, the results analysis and discussion are not scientific and logical, so there are several points which should be further improved and corrected. Therefore, this manuscript should be totally revised, restructured, and resubmitted again for reviewing.

  1. The citric and acetic used as the ammonia absorbents for ammonia recovery were reported from the technological and economical perspective in the paper [Z. Jamaludin, S. Rollings-Scattergood, K. Lutes, C. Vaneeckhaute, Evaluation of sustainable scrubbing agents for ammonia recovery from anaerobic digestate, Bioresource Technology 270 (2018) 596-602]. Additionally, the feasibility of cold water, sulfuric, phosphoric, and nitric used as the ammonia absorbent were reported elsewhere. The authors should clarify the novelty in this manuscript.
  2. The abstract is ambiguous because of its poor logic. Please re-write it. For example, the introduction of the background section is too redundant, and the summary of the experimental conclusion data is too short. In addition, the results of the economic analysis are not shown in the abstract section, and only conclusive language statements are included, which is not convincing enough.
  3. Keywords must be reconsidered. For example, “e-PTFE”, I do not think it is a suitable keyword for this paper. The present study was concentrated on the ammonia capture performance of different ammonia absorbents.
  4. Why the experimental conditions with pH value of 8 and feed temperature of 25 °C were chosen?

In this condition, the free ammonia concentration is relatively low. What’s the ammonia removal efficiency after 7-day experiment?

  1. The concrete influence of different ammonia absorbents on ammonia absorbent performance in the 7-day GPM process should be descripted clearly, not only a total experiment results after the 7-day operation. For example, did the absorption rates of different absorbents change over the 7-day trials? Can the absorbent absorb all the ammonia nitrogen in the feed liquid according to your experimental design?
  2. Please supplement the data in Table 2 and Fig. 1, and complete the changes of different parameters in the 7-day process.
  3. line 232, there were significant differences between sulfuric acid and phosphoric acid. Why?
  4. The language should be further polished.

Author Response

Reviewer #1

Comments and Suggestions for Authors

In the present study, the author reported the ammonia recovery performance of eight different capture solutions at two temperatures (25 and 2 °C) using suspended expanded GPMs in a laboratory-scale experimental setup. Additionally, this manuscript also provided an economic evaluation for different ammonia capture solutions. However, the results analysis and discussion are not scientific and logical, so there are several points which should be further improved and corrected. Therefore, this manuscript should be totally revised, restructured, and resubmitted again for reviewing.

Q1. The citric and acetic used as the ammonia absorbents for ammonia recovery were reported from the technological and economical perspective in the paper [Z. Jamaludin, S. Rollings- Scattergood, K. Lutes, C. Vaneeckhaute, Evaluation of sustainable scrubbing agents for ammonia recovery from anaerobic digestate, Bioresource Technology 270 (2018) 596-602]. Additionally, the feasibility of cold water, sulfuric, phosphoric, and nitric used as the ammonia absorbent were reported elsewhere. The authors should clarify the novelty in this manuscript.

Response: As indicated by the Reviewer, the solutions under study have been previously used for ammonia capture in other investigations (in fact, we cited the suggested study by Jamaludin et al. once in the introduction and four times in the discussion section). Nonetheless, they were tested in connection with other TAN recovery technologies, viz. air stripping systems, liquid-liquid membrane contactors, hollow fiber membrane contactors, scrubbing systems, flow-electrode capacitive deionization systems, and capacitive membrane extraction systems. Given that in this study the performance of the solutions was evaluated on a different type of system, based on suspended gas-permeable membranes (at different temperatures), the novelty is not compromised. A comment to clarify this matter has been included in the revised introduction section (which has been shortened as per Reviewer #2’s request): “[…] Some authors have explored the possibility of using other mineral acids (nitric acid, phosphoric acid, acid mixtures, etc.) [34-37] for NH3 removal from tertiary effluents, wastewater or human urine, while other researchers have used organic acids, carbonic acid, or water [38-40] for NH3 recovery from anaerobic sewage sludge digestate, landfill leachate, or wastewater. Nonetheless, the aforementioned studies explored those alternative capture solutions in air stripping systems, liquid-liquid membrane contactors, hollow fiber membrane contactors, scrubbing systems or flow-electrode capacitive deionization systems, or capacitive membrane extraction systems (CapAmm), not on air-suspended GPM systems. […]”

Q2. The abstract is ambiguous because of its poor logic. Please re-write it. For example, the introduction of the background section is too redundant, and the summary of the experimental conclusion data is too short. In addition, the results of the economic analysis are not shown in the abstract section, and only conclusive language statements are included, which is not convincing enough.

Response: The abstract has been rewritten taking into consideration the Reviewer’s suggestions (the background has been shortened, results have been expanded to include the economic analysis, and the final statement has been softened).

Q3. Keywords must be reconsidered. For example, “e-PTFE”, I do not think it is a suitable keyword for this paper. The present study was concentrated on the ammonia capture performance of different ammonia absorbents.

Response: The keywords have been updated: ‘e-PTFE’ has been removed and ‘ammonia adsorbents’ has been included.

Q4. Why the experimental conditions with pH value of 8 and feed temperature of 25 °C were chosen?

Response: The experiments carried out in this article were performed in conditions that intend to replicate those found in real farm applications. Hence, the synthetic solution was prepared at a pH similar to those of real emitting sources (pig slurry, chicken manure, laying hen manure, etc.) that can be found on farms [see, for instance, https://doi.org/10.1016/j.watres.2020.116789, Table 1; https//doi.org/10.1016/j.wasman.2015.01.021; and https://doi.org/10.2166/wst.2017.116]. A clarification and the references above have been included in the revised version.

Concerning the 25 ºC feed temperature, it is the usual setpoint temperature in many Spanish farms [see, for instance, https://doi.org/10.3390/agronomy10010107] and would be the approximate temperature at which the NH3 emitting source would be in real farm conditions.

Q5. In this condition, the free ammonia concentration is relatively low. What’s the ammonia removal efficiency after 7-day experiment?

Response: Concerning the first part comment, the average ammonia emission over the 7 days of the experiment was close to 53% in all the tests (please note that the NH3 emitting solution was always at 25 ºC, regardless of whether the capture solution was at 25 ºC or 2 ºC), so the free ammonia concentration was ca. 3200 ppm. A clarification has been included in subsection 3.2.4. About the second part of the comment, the ammonia removal efficiency at the end of the experiment, considering the amount of ammonia finally recovered in the trapping solution compared to the initial ammonia concentration in the emitting solution (not vs. the emitted NH3), has now been included in Table 2 (in addition to the capture efficiency). Subsection 2.4 has also been updated accordingly to include this parameter.

Q6. The concrete influence of different ammonia absorbents on ammonia absorbent performance in the 7-day GPM process should be descripted clearly, not only a total experiment results after the 7-day operation. For example, did the absorption rates of different absorbents change over the 7-day trials? Can the absorbent absorb all the ammonia nitrogen in the feed liquid according to your experimental design?

Response: A new figure with the (linear) evolution of the capture process for each adsorbent and temperature has been included and briefly discussed in a new subsection in the revised manuscript (subsection 3.2.1), to address the Reviewer’s first query. Please note that in previous experiments with sulfuric acid as the trapping solution, which were conducted over 14 days [10.3390/membranes11070538] and over 240 days [DOI: 10.3390/membranes11110859], we also observed that the evolution of the amount of TAN captured in the capture solution was linear, with quite constant capture rates.

Concerning the second question, as it is now shown in the updated Table 2 and discussed in the revised subsection 3.2.4, the adsorbents could capture most of the emitted NH3 without reaching saturation (at least in short experiments), but almost half of the TAN remained in the emitting solution (because the ammonia nitrogen which is not-emitted from the feed liquid cannot be absorbed with a suspended GPM system). Values from other studies have been included for comparison purposes.

Q7. Please supplement the data in Table 2 and Fig. 1, and complete the changes of different parameters in the 7-day process.

Response: The Reviewer’s request has been addressed (please see the response to Q6 above).

Q8. line 232, there were significant differences between sulfuric acid and phosphoric acid. Why? Response: A possible explanation has now been included in the manuscript, explaining that ammonium phosphate is less soluble in water than ammonium sulfate, with solubility values of

42.9 kg·L-1 for [(NH4)2PO4] and 70.6 for [(NH4)2SO4] at 0 °C, and of 68.6 vs. 75.4 kg·L-1 at 20 ºC, respectively. Therefore, better performance of sulfuric acid in terms of NH3 capture than that of phosphoric acid may be expected. A supporting reference has been included.

Q9. The language should be further polished.

Response: The manuscript has been reviewed by a colleague and revised to improve readability. The final version has been checked with GrammarlyPro.

Reviewer 2 Report

The authors presented some nice work on using different absorbents in membranes to recover ammonia. It is not very novel or exciting but the manuscript told a good story. There are several points that I believe the authors can work on. 

1. The introduction is unnecessarily long and disorganized. There is not a logic line in it. Please try to put some paragraphs together and be concise. 

2. The 7 days study is good however, it would be helpful to analyze the membranes changes after the 7 days working. Recommend to perform some characteristics analysis. 

3. A good story is always a full story, I do not understand the Future Lines of Research session here. Either take it out, or execute it and write it in this paper, the current status is not good for a journal article. 

Author Response

Reviewer #2

 The authors presented some nice work on using different absorbents in membranes to recover ammonia. It is not very novel or exciting but the manuscript told a good story. There are several points that I believe the authors can work on.

Q1. The introduction is unnecessarily long and disorganized. There is not a logic line in it. Please try to put some paragraphs together and be concise.

Response: The introduction has been modified according to the Reviewer's indications. Apart from putting paragraphs together, it has been shortened from 896 to 766 words (even though new information has been included in response to the other two Reviewers’ requests).

Q2. The 7 days study is good however, it would be helpful to analyze the membranes changes after the 7 days working. Recommend to perform some characteristics analysis.

Response: In this study, new membranes were used for each 7-day test. Given that they were not reused, a morphological analysis of the membranes was not conducted. This point has now been clarified in subsection 2.2 (“[…] Given that the membranes were not reused, a morphological analysis of the membranes was not conducted after the 7-day experiments.”). However, it is worth noting that the deterioration of the gas-permeable membrane in suspended systems is much slower than in submerged systems, so long use periods are required to observe differences (no changes may be expected in week-long experiments). In our experience, in e-PTFE membranes that were analyzed after ‘long-term’ pilot-scale tests (which lasted for approximately a year), salt fouling occurred, and periodic cleaning and monitoring of the characteristics of the membrane would be advisable.

Q3. A good story is always a full story, I do not understand the Future Lines of Research session here. Either take it out, or execute it and write it in this paper, the current status is not good for a journal article.

Response: The future lines or research section has been deleted, following the Reviewer’s recommendation.

Reviewer 3 Report

This paper presents on the effects of eight capture solution variations at two temperatures (25 degC and 2 degC) at a laboratory scale using suspended expended polytetrafluoroehthylene GPMs. It is found that water, phosphoric acid, and citric acid are the ideal NH3 adsorbents in this work. This is an interesting work and few comments required to be addressed by the authors of this paper to improve the quality of this manuscript.

The comments are as follows:

  1. Introduction section: How can this work benefits to both researchers and scientists? Please elaborate.
  2. Section 2.2.: Please mentioned how many times each experimental run had been repeated.
  3. Section 2.5: Please explain why one-way ANOVA method is used in this research. 

Author Response

Reviewer #3

 This paper presents on the effects of eight capture solution variations at two temperatures (25 degC and 2 degC) at a laboratory scale using suspended expended polytetrafluoroehthylene GPMs. It is found that water, phosphoric acid, and citric acid are the ideal NH3 adsorbents in this work. This is an interesting work and few comments required to be addressed by the authors of this paper to improve the quality of this manuscript.

The comments are as follows:

Q1. Introduction section: How can this work benefits to both researchers and scientists? Please elaborate.

Response: The requested information has been included at the end of the introductory section in the revised manuscript: “[…] The reported technical and preliminary economic evaluation may not only be useful to researchers working on more sustainable membrane-based systems, but also to livestock farm owners and agricultural operators interested in producing certified organic fertilizers (which is not possible if mineral acids are used).”

Q2. Section 2.2.: Please mentioned how many times each experimental run had been repeated. Response: The information requested by the Reviewer appeared at the end of subsection 2.2 ("For each combination of capture solution and temperature, three replicates were performed over a period of 7 days"), but we have replaced “replicates” with “runs”, in agreement with the Reviewer’s wording.

Q3. Section 2.5: Please explain why one-way ANOVA method is used in this research. Response: We would like to thank the Reviewer for bringing this mistake to our attention. Indeed, a two-way ANOVA method was used, not a one-way ANOVA (given that we needed to study the relationship between a quantitative dependent variable and two qualitative independent variables). The two-way ANOVA allows us to study how each of the factors (type of NH3 extraction solution and temperature) influenced the dependent variable (ammonia capture), as well as the influence of the combinations that can occur between them. The mistake has been corrected in subsection 2.5 of the revised version of the manuscript.
